# A Critical Review of the Study of Neuroprotective Diets to Reduce Cognitive Decline

**DOI:** 10.3390/nu13072264

**Published:** 2021-06-30

**Authors:** Sally C. Duplantier, Christopher D. Gardner

**Affiliations:** 1The USC Leonard School of Gerontology, University of Southern California, 3715 McClintock Ave., Los Angeles, CA 90089, USA; sduplant@usc.edu; 2Stanford Prevention Research Center, Department of Medicine, School of Medicine, Stanford University, Stanford, CA 94305, USA

**Keywords:** Mediterranean, DASH, MIND, dietary pattern, cognitive decline, cognition, Alzheimer’s, dementia

## Abstract

Alzheimer’s disease (AD) and other dementias are now the seventh leading cause of death in the world and are projected to affect 115.4 million people by 2050. Delaying the onset of AD by just five years is estimated to reduce the cost and prevalence of the disease by half. There is no cure for AD nor any drug therapies to halt its progression once the disease begins. Lifestyle choices including diet are being seen as a viable complementary therapy to reduce cognitive decline, the hallmark of AD. Mediterranean, DASH (Dietary Approaches to Stop Hypertension), and MIND (Mediterranean-DASH Intervention for Neurodegenerative Delay) diets have biological mechanisms supporting their potential neuroprotective benefits, but the findings of study outcomes about these benefits have been inconsistent. This paper analyzed five Randomized Clinical Trials (RCTs) (from 2000 to 2021) and 27 observational studies (from 2010 to 2021) focused on the link between cognitive health and the Mediterranean/DASH/MIND diets to identify gaps and challenges that could lead to inconsistent results. These include a lack of accuracy in assessing food intake, multiple dietary pattern scoring systems, a shifting metric among studies focused on the Mediterranean diet, a lack of standards in the tools used to assess cognitive decline, and studies that were underpowered or had follow-up periods too short to detect cognitive change. Insights from these gaps and challenges are summarized in recommendations for future RCTs, including both pragmatic and explanatory RCTs.

## 1. Introduction

Alzheimer’s disease (AD) and other dementias are now the seventh leading cause of death in the world [1]. AD is a disease with no cures or therapies to reduce its progression and symptoms once they start [2]. Currently, more than six million Americans are living with AD, and this number is expected to reach nearly 13 million by 2050 at an annual cost of $1.1 trillion [3]. Globally, AD and other dementias are projected to affect over 115 million people by 2050 [4].

There have been no new drug therapies for AD in more than a decade, and current therapies only manage disease symptoms but do not halt, slow, or reverse the disease itself [5]. Lifestyle choices, including diet and nutrition, are being examined as a complementary method to delay the onset of AD. Early therapeutic action is needed to address biological processes that may already be compromised, and before AD progresses from a state of mild cognitive impairment (MCI) to cognitive failure [6]. By 2050, a five-year delay in the onset of AD could reduce the prevalence and cost of this disease by 50% and dramatically decrease the burden of AD on patients, their families, and the health care system [7].

AD involves the intraneuronal formation of abnormal tau protein and extracellular deposition of beta-amyloid protein in the central nervous system [8]. The pathology is a continuum vs. a discrete disease state, with tau tangles and beta-amyloid protein plaques forming years or decades before symptoms appear. Over time, AD-related lesions develop at the sites of these tangles and plaques and spread to previously uninvolved areas of the brain [8]. Knowing the pathophysiology and targets, a nutritional approach would focus on dietary components and nutrients with the potential to act on those targets.

Dietary intake of antioxidants, B vitamins, polyphenols, and polyunsaturated fatty acids (PUFAs), found in vegetables, fruits, nuts, whole grains, legumes, nuts, seeds, and red wine, may play a role in reducing the risk of AD or delaying its onset. These nutrients inhibit the formation of beta-amyloid peptide oligomers and fibers associated with AD [9]. Monounsaturated fats (MUFAs) found in found in olive oil, nuts, and seeds play an important role in inducing autophagy, a deficit which is associated with early-stage AD [10]. Autophagy contributes to the generation and metabolism of beta-amyloid and the assembling of tau, so its malfunction may lead to the progression of AD [11]. Docosahexaenoic acid (DHA), found in fatty fish such as salmon, has been shown to reduce levels of tau protein in animal models and increase a specific protein, LR11, which is important to brain health. A deficit of LR11 is associated with increased beta-amyloid production [12]. DHA is also a key component of the brain lipids, and an adequate intake of DHA may help maintain neuronal integrity and function through modification of the gene expressions associated with neurogenesis and neuronal function [13,14]. Antioxidants, B vitamins, polyphenols, PUFAs, MUFAs, and DHA also reduce the risk of systemic inflammation and oxidative stress that are strongly linked to AD [15,16].

The Mediterranean (Medi), DASH (Dietary Approaches to Stop Hypertension), and MIND (Mediterranean-DASH Intervention for Neurodegenerative Delay) diets provide a source of antioxidants, B vitamins, polyphenols, PUFAs, MUFAs, and DHA. The Medi diet is a dietary pattern associated with the lifestyle of the Mediterranean people. The DASH diet was originally designed to help treat or prevent hypertension. The MIND diet was developed specifically for its neuroprotective effects by a team at Rush Medical Center in 2015. The Medi, DASH, and MIND diets are primarily plant-based and include vegetables, fruits, nuts, whole grains, legumes, nuts and seeds, healthy fats, fish, and poultry that are rich in these nutrients. The Medi, DASH, and MIND diets minimize red meat, sweets, processed foods, and sugary drinks that are often associated with Western diets [17]. The Western diet is linked to an increased risk of AD, although the underlying mechanisms are not fully understood. In mouse models, the Western diet accelerated brain inflammation and the production of beta-amyloid protein [18]. Additionally, the Medi, DASH, and MIND diets reduce middle-age risk factors for AD, including vascular disease, hypertension, stroke, diabetes, and obesity [19,20,21,22,23,24,25]. In particular, vascular disease risk factors, especially in mid-life, are associated with the build-up of beta-amyloid in the brain [26,27]. It is estimated that 40% of dementia cases could be prevented or delayed by targeting modifiable lifestyle risk factors [28].

A list of the Medi/DASH/MIND dietary components and their potential neuroprotective benefits is presented in Table 1. Taken individually, these dietary components may promote cognitive health. Taken together, these components form patterns, and emerging research suggests that the synergistic effect of healthy dietary patterns may affect both the physiological processes and signaling pathways associated with cognitive function and decline [29,30,31]. The Medi diet is associated with reduced risk of cognitive decline in Mediterranean populations [32,33,34] and non-Mediterranean populations [35,36,37]. The DASH diet has also been associated with better cognitive function [38] and reduced risk of AD [17]. The MIND diet combines elements of the Medi and DASH diets and is associated with reduced risk of cognitive impairment [17,39,40]. While the Medi/DASH/MIND diets are all plant-based, there are key differences in the types of food and recommended amounts, which are reflected in Table 2.

Based on current research, it is unclear whether the Medi, DASH, or MIND diet is superior for preventing or reducing cognitive decline due to several factors. Most studies to date have been observational, lacking the rigor of RCTs [58] (p. 5). Within these studies, there is lack of consistency in the measures related both to dietary intake and cognitive change [41]. Despite the limitations in measurement tools and the inconsistencies in study outcomes mentioned, there is sufficient evidence of the potential neuroprotective benefits of all three dietary patterns [59] to warrant further investigation, especially given that the risks are so low and the stakes are so high. A treatment that delays the onset or progression of AD would have a significant impact on improving the quality of life for individuals and their families while simultaneously reducing caregiver burden and healthcare costs [3].

This paper provides a critical review of the current literature related to neuroprotective dietary patterns and cognitive decline. It identifies gaps in the design of current studies which have led to inconsistent findings within the literature reviewed and challenges in filling those gaps. This paper also recommends solutions for more robust study designs in the future. These solutions include gold-standard explanatory Randomized Clinical Trials which demonstrate efficacy, as well as pragmatic RCTs which demonstrate the effectiveness of interventions in real-life environments such as clinical practices or community settings.

## 2. Materials and Methods

For this review, we searched the PubMed and InfoPsych databases using the key search terms “Mediterranean or DASH or MIND and cognitive decline or cognition or Alzheimer’s or dementia”. To narrow the volume of publications, the terms “and systematic review or meta-analysis” were added. The search yielded three systematic reviews [2,30,59]. Combined, these three systematic reviews provided 140 papers that were identified from these databases: PubMed, Medline, Cochrane Library, Embase, Google Scholar, Web of Science, and Scopus. An additional search performed to find relevant papers published since 2019 yielded one additional RCT [2], which is still underway.

Included are all peer reviewed RCTs from 2000 and all observational studies from 2010 on the Mediterranean, DASH, and/or MIND diets. Excluded are studies on similar dietary patterns such as the Anti-Inflammatory Diet or the Prudent Healthy Diet because data are more limited for these [41]. From the studies that remained, 18 duplicates were eliminated, yielding a total of five RCTs and 27 observational studies. Six of the 27 observational studies included more than one dietary pattern (e.g., Mediterranean, DASH, and/or MIND diets were compared in one paper). This created a total of 35 dietary analyses based on observational studies, in addition to the five RCTS, for a total of 40 dietary analyses. The analysis of outcomes is based on our understanding of the primary hypothesis of each study.

## 3. Gaps and Challenges in Current Research

This paper has identified the potential ways in which neuroprotective diets may contribute to cognitive health, while recognizing that research to date has been inconsistent, and in some cases, conflicting. As examples, this paper highlights the 32 studies reviewed on the association between the Medi/DASH/MIND dietary patterns and cognition or cognitive decline. These papers cover five RCTs and 27 observational studies (Figure 1). Because some studies included more than one dietary pattern, the review includes results for a total of 40 dietary analyses. Of the five RCTs, four included a Medi diet arm, with two of these reporting a positive impact between the Medi diet and cognitive health [33,34] and one reporting a null effect [60]. The fourth Medi RCT [61] reported mixed results; there were no differences between either of the two intervention groups or the control group on three measures of cognitive function. For a fourth measure, which involved the task with the greatest processing load, the two intervention groups adhering to the Medi diet did significantly worse (*p* < 0.001) than the control group. The fifth RCT, testing a DASH diet [62], also had mixed results. The intervention with the DASH diet alone (without exercise) showed improvement in psychomotor speed but not in executive function-memory learning. Within the observational studies representing 35 analyses of Medi/DASH/MIND, there were 21 positive associations between diet and cognition, nine null findings, and five studies with mixed outcomes. Details about the RCTs and observational studies can be found in Table 3.

### 3.1. Challenges in Measuring Dietary Impact on Cognition

This section addresses the gaps and challenges of identifying the link between dietary patterns and cognitive change. These gaps and challenges contribute to the inconsistencies of study results to date. Some of these challenges are unique to the study of diet and cognition, and other challenges are common to any clinical trial or observational study but play a significant role in accurately assessing cognitive decline.

#### 3.1.1. Dietary Intake Measures

The key to analyzing the link between neuroprotective diets and cognitive health is the accuracy of dietary intake. Inaccuracy here can lead to inconsistencies in results across studies. Common tools include food frequency questionnaires (FFQs), 24-h dietary recalls (24HRs), or food diaries. Dietary intake measures varied widely in the five RCTs we reviewed. One study used an FFQ, two studies used a combination of an FFQ and questionnaire, one study used dietary diaries, and the final RCT did not specify the dietary intake method. Within the observational studies, most (*N* = 23) incorporated FFQs or semi-quantitative FFQs, several (*N* = 3) used 24HRs, and one used food diaries.

All dietary intake tools that involve self-report have inherent limitations which contribute to inaccuracies. For FFQs, used predominantly in large-scale epidemiologic studies, these limitations include inaccuracy related to portion size [83], lack of generalizability to multi-racial/ethnic cohorts [84], and the addition of non-standard items unique to the Medi, DASH, or MIND diets which may not exist on the FFQ and contribute to misclassification [85]. Limitations to 24HRs are the cost and logistics of the data [86] and the inability to account for day-to-day variation in diet. Limitations to food diaries include cost, logistics, and high respondent burden [86,87]. To improve the accuracy of estimated individual intake, research supports the use of a combination of tools such as the FFQ and multiple 24HRs [88].

#### 3.1.2. Dietary Pattern Metrics

A second challenge in accurately assessing dietary intake is that multiple tools have been created for scoring adherence to the same dietary pattern. Radd-Vagenas et al. [89] identify 30 different Mediterranean Diet scoring (MDS) systems, and there are at least 10 different DASH scoring systems [85]. The two MDS systems most frequently used are the original scoring system by Trichopoulou et al. [54] and an alternate by Panagiotakos et al. [90]. The Trichopoulou et al. [54] score developed in 2003 ranges from 0 (minimum adherence) to 9 (maximum adherence) based on sex-specific median intakes of nine components. The Panagiotakos et al. [90] score ranges from 0 to 55 based on 11 components. None of the four Medi RCTs reviewed here used either the original or alternate scoring system. Two of the Medi RCTs used their own 14 item questionnaire to score adherence [33,34]. One Medi RCT created their own scoring system for an Australian population based on the food content of 15 Medi diets and the nutrient content of eight [60]. The final Medi RCT did not identify a specific Medi scoring system but instead provided participants with guidelines to increase vegetables, fruits, and oily fish, reduce dietary fat to 30% of energy, and substitute predominantly monounsaturated fats for saturated fats [61]. Within the Medi observational studies reviewed, most used the original method (*N* = 18), some used the alternate method (*N* = 5), and one study used both.

The MDS scoring metrics are uniquely problematic as they are typically based on median cut points for intake of food groups, which differ from one population to another [91]. For example, in Greece, the median intake of vegetables is six servings a day, and the median intake of fruits is three servings [55,91]. An individual will receive a 0 or 1 score based on whether they were below or above the median. In contrast, the US median intake of vegetables is three servings a day and the median intake of fruits is two servings [92]. This creates a floating metric of intake across studies. Because the intake of vegetables is higher in Mediterranean countries, there is the potential that studies conducted here will show more positive associations with this dietary pattern and cognitive health. In the four Medi diet RCTs we analyzed, the two conducted in Mediterranean countries showed positive results, and the two in non-Mediterranean countries showed null results. Within the Medi diet observational studies, the three conducted in Mediterranean countries showed positive results. Within non-Mediterranean countries, 10 had positive results, eight had null results, and three had mixed results. An additional challenge within non-Mediterranean countries is the country-specific interpretation of the traditional scoring systems. To reflect the eating pattern of Swedes, the ULSAM observational study of the Medi diet replaced monounsaturated fatty acids with polyunsaturated fatty acids, excluded nuts and seeds, and combined potatoes with whole grains, even though it used the 0–9 MDS scoring system [73]. Similar adjustments were made in observational studies for Chinese populations [37], Australian populations [67,68], and Nordic populations [76].

The DASH scoring metrics also vary widely, with more than 10 being used [85], although the most common were developed by Fung et al. [56] and Folsom et al. [93]. The Fung et al. [56] score is based on eight dietary components, each of which receives a score of 0 to 1 based on quintile intake, with a total score between 8–40. The Folsom et al. [93] score is based on pre-defined cut-offs of 11 components, with a total score between 0–11. Both were used in the observational studies that were analyzed (Fung et al. score *N* = 3, Folsom et al. score *N* = 1). The dietary pattern in the one RCT focused on the DASH diet did not specify either of these dietary pattern scores but measured changes according to the Healthy Eating Index [62].

In contrast to the challenges with scoring the Mediterranean diet, which is based on median intake, or the multiple scoring systems for the DASH diet, the MIND diet is easier to score. Currently, there is only one system, and it is based on absolute vs. relative levels of intake [91]. It assigns one point for the specified consumption per day or week of 10 brain healthy foods (whole grains, leafy greens, other vegetables, berries, fish, poultry, beans, olive oil, nuts, and wine) and assigns one point for consuming less than the maximum specific portion per day or week of five unhealthy brain foods (red meat, fast/fried food, butter/margarine, cheese, and pastries/sweets [39]. The 2015 MAP observational study, Table 1 [17] provides an example of how all three dietary patterns may be scored in a single study. 

#### 3.1.3. Cognitive Assessment Measures

In the recent book published by the National Academy of Sciences, *Preventing Cognitive Decline: A Way Forward*, the authors identify a lack of consistent cognitive outcome measures across current and recent trials to be a major challenge in understanding the effects of lifestyle interventions such as diet on cognition, making it impossible to pool results across studies [58] (pp. 84–85). For meaningful outcomes, it is recommended that the measures used to assess cognitive decline reflect the latest diagnostic guidelines for assessing AD. However, those guidelines are shifting, and research methods have not kept pace. The Alzheimer’s Disease Diagnostic Guidelines, established more than 30 years ago [94], were updated in 2011 [95] to better assess the disease at its earliest stages. Assessment now requires a battery of tests across cognitive domains (e.g., working memory, semantic memory, visuospatial ability, and perceptual speed) in addition to episodic memory, where the first major outward symptoms of dementia are seen [95]. In addition, the new guidelines recommend that biomarkers such as structural magnetic resonance imaging (MRI) be used in research-based studies to identify disease presence, even though an MRI cannot confirm dementia with certainty [95,96].

Within the RCTs reviewed here, four used a battery of neuropsychological tests, but the 2013 PREDIMED-NAVARRA trial [33] used only the Mini-Mental State Examination (MMSE) and Clock Drawing Test (CDT). Both are considered crude measures of assessing cognitive decline which lack sensitivity to small changes [97] with the potential to under-report cognitive change. In addition, the 2013 PREDIMED trial failed to administer these tests at baseline, so it could only analyze cognitive function, not change in cognition, over the 6.5 years of the study [33]. Within the observational studies that were reviewed, researchers used a wide range and number of neurological assessments (*N* = 1–21). Similar to the RCTs, some (*N* = 5) used only the MMSE to assess cognitive change. Others used simplified measures such as the Subjective Cognitive Function (SCF) test, with only six yes/no questions based upon participants’ perception of their current cognitive functioning [64] or the Six Item Screener, with just six questions that result in a rating of “impairment likely” or “impairment less likely” [81]. These assessments fall short in analyzing the complexity of cognitive change, making it more difficult to interpret results across studies. In contrast, a battery of neurological assessments provides a more complete picture of the effects of diet on specific cognitive functions, although a downside to testing batteries is that they often provide mixed results, leading to challenges in interpretation of overall findings. These mixed results occurred in two of the RCTs studied and four of the observational studies. Examples of commonly used tests in neuropsychological test batteries are found in Table 4. Even though biomarkers such as structural magnetic resonance imaging (MRI) are now part of the standard guidelines for diagnosing AD [95], only the PIVUS observational study in Sweden [80] used brain imaging in addition to neuropsychological tests.

### 3.2. Gaps Common to Clinical Trials and Observational Studies

#### 3.2.1. Length of Follow-Up

For a study to generate substantive findings, it must be long enough to capture at least a minimum number of events or changes in measures that involve continuous outcomes. There is no gold standard for the number of years required to measure cognitive decline, but it is difficult to detect in periods of less than a year [58] (p. 81). Within the five RCTs reviewed, three had durations of less than a year. The MedLey study (null finding) was six months [60]; the ENCORE study (mixed findings) was four months [62]; and the Wardle et al. study (mixed findings) was 12 weeks [61]. Insufficient follow-up periods lead to spurious results.

#### 3.2.2. Attrition

A challenge in longer studies is participant attrition. The two PREDIMED-NAVARRA RCT studies [33,34] had longer follow-up periods than the three already cited (6.5 and 4.1 years, respectively). Both found a positive link between the Medi diet and cognition, although these studies suffered from a common challenge of longer trials: high attrition. Because of high drop-out rates in both studies, the Agency for Healthcare Research and Quality considered both to be at high risk for survivor bias [58] (p. 113).

#### 3.2.3. Statistical Power and Sample Size

Ideally, studies are designed to detect a meaningful minimum difference (if there is one), and to provide confidence that a null finding is due to a lack of an effect when that is the true relationship, and not an error in study design (e.g., inadequate sample size, high attrition). The statistical “power” of a study is largely determined by the variance in study variables and sample size. Sample size is especially important in the study of cognitive change because the subset of participants who develop cognitive decline is relatively small and drawing conclusions from a small subset can be misleading. This was the challenge of the PATH Through Life observational study [65] which analyzed the impact of the Medi diet on the cognition of 1528 men for four years and found that monounsaturated fatty acids (MUFAs), combined with excessive calorie intake, were predictive of MCI. This finding on monounsaturated fats is the oppositive of dozens of studies reporting neuroprotective benefits of MUFAs [29,33,34]. However, the conclusions of the PATH Through Life study are based on a small subset of the study population who developed MCI or dementia: 10 participants were diagnosed with MCI (0.6%); 19 had a Clinical Dementia Rating of 0.5 (1.2%); and 37 were diagnosed with mild cognitive disorder (2.4%) [65].

## 4. Beyond Study Limitations to Practical Recommendations

While past studies have limitations, it is important to learn from them and take advantage of new tools and more robust design methods so that these neuroprotective dietary patterns can be introduced to more individuals and a broader array of researchers. This section provides two distinct ways to accelerate future research through RCTs in the form of both pragmatic trials and explanatory trials.

### 4.1. The Role of Pragmatic Trials in the Study of Neuroprotective Diets

Clinical trials, the gold standard within the medical sciences community to test and evaluate interventions, fall into two broad categories: explanatory and pragmatic. Many trials have features of both designs [99,100]. Explanatory trials are designed to test whether an intervention works under optimal conditions (efficacy), while a pragmatic trial evaluates the effectiveness of an intervention in real-life practical conditions, such as a clinical care setting. Explanatory trials carefully control the inclusion and exclusion criteria of participants while pragmatic trials have broader inclusion criteria that is more representative of the general population [101].

In its recommendations for public health interventions to prevent or delay cognitive impairment, the NAS identifies pragmatic trials as one of the top methodological strategies to get dementia-related interventions into broad, routine clinical practices or community settings [58] (p. 88). An example of a pragmatic trial is AgeWell.de, a WW FINGERS trial funded by the Alzheimer’s Association [100]. A major goal of the WW FINGERS research is to get interventions to as large of a population as possible in real-world settings so that they can be understood, refined, and applied by practitioners [102,103]. The AgeWell.de study is being conducted by primary care physicians within their offices and includes a wide range of older adults who are part of their physician’s general practice.

### 4.2. New Standards in Explanatory RCTs for Neuroprotective Diets

This section provides specific recommendations for future explanatory RCTs, drawing upon an example of an RCT underway at Rush Medical Center [2]. The study is a three-year RCT designed to test the effects of the MIND dietary pattern on the cognitive function of 604 men and women, 65–84 years old, cognitively unimpaired, and overweight, with suboptimal diets. Both the intervention group and control group will be prescribed mild caloric restriction for weight loss.

#### 4.2.1. Dietary Intake Measures

Use a combination of 24HRs and FFQs. The Rush RCT [2] combines both measures and applies these at baseline and at months 6, 12, 24, and 36. This combination of both measures across all test intervals is designed to allow for more accurate intake of dietary components. In the studies examined in this paper, the FFQs and semi-quantitative FFQs varied widely across the 32 studies reviewed in this paper. Researchers should select an FFQ that has been validated within the population to be studied. For RCTs with insufficient funding for staff administered 24-h recalls, an option is the Automated Self-Administered 24-h (ASA24^®^) Dietary Assessment Tool [104]. This is a free, web-based tool developed in 2009 in conjunction with the NIH. It enables multiple, automatically coded, self-administered 24-h diet recalls and/or single or multi-day food records and has been used in more than 6000 studies as of January 2020.

#### 4.2.2. Dietary Pattern Metrics

Measure absolute rather than relative intake levels. There are advantages to scoring dietary patterns on an absolute intake of food, which is the norm for the MIND diet, rather than on relative cutoffs based on within population levels of intake, as in the Medi diet. Absolute intake makes it easier to interpret results across different studies because they share a common metric. It is also easier to understand when presented as servings per day [91]. All three dietary patterns discussed can be scored in absolute values vs. relative (Table 4). To pool results across studies, the servings per day should be consistent across populations, not altered for median intake.

#### 4.2.3. Cognitive Assessment Measures

Use a neuropsychological test battery combined with a biomarker test (at least in a subset of the study population). These recommendations reflect the latest Alzheimer’s Disease Diagnostic Guidelines [95] and take advantage of new tools to measure cognitive change. A recommended tool for screening is the Montreal Cognitive Assessment (MoCA) [37]. Compared to the MMSE, which has been cited in many of the studies reviewed here, the MoCA shows less susceptibility to both cultural and educational biases and is free for clinical use [90]. The MoCA also has the advantage of testing multiple cognitive domains with an easy and single scoring system [96]. For the neuropsychological testing, an option is the NIH Toolbox (nihtoolbox.org). This set of instruments was developed within the past decade and designed to establish a standard approach to measuring cognitive, emotional, sensory, and motor function, allowing data across research studies to be pooled. Additionally, at least a subset of participants should have MRI scans at the beginning and end of the intervention period to assess change in measures of total brain volume along with other measures of brain macro-and micro-structural integrity [2].

#### 4.2.4. Length of Follow-Up

Longer is better but allow more than one year. As stated earlier, there is no gold standard for the number of years needed to assess cognitive change although various sources offer a range. The NAS acknowledges that one year is too short of a duration and recommends 5 to 10 years [58] (p. 82). A systematic review relating to change point studies on accelerated decline in cognitive and neurological outcomes found that change points related to MCI were 3–7 years prior to diagnosis, and change points related to AD and other forms of dementia were 1–11 years prior to diagnosis [105]. The two Medi diet RCTs with positive findings were 6.5 years and 4.1 years, respectively [33,34]. The Rush RCT [2] is designed to include a follow-up of more than three years, although the study authors recognize this relatively short timeframe may be a limitation.

#### 4.2.5. Adherence

Establish measures to achieve and assess adherence. Most RCTs have questionable long-term adherence to the dietary intervention selected [106]. To address this, there are two separate components to consider: achieving adherence and assessing adherence. In many RCTs, these aspects of study design and conduct are overlooked or minimized, or adherence is confused with attending a class vs. eating the prescribed foods. The Rush RCT [2] applies a rigorous and consistent method to support dietary adherence. This includes an initial three- to four-week run-in period to assess compliance, combined with frequent telephone/email communications with a nutritional case manager, personalized diet plans, compliance aids including a food tracking phone app, group cooking classes and buddy systems, and frequent weight monitoring and check-ins. Assessment of adherence in that study is conducted using both FFQs and 24HRs, as described previously. These approaches are aligned with maximizing both the level of adherence achieved and the ability to report adherence achievement to allow for more effectively critical evaluation.

#### 4.2.6. Attrition

Help participants feel valued and maximize their engagement. Selective attrition is a potential problem in longer-running RCTs (as well as longitudinal studies). If the people who continue to participate differ from their drop-out counterparts with respect to key characteristics, the remaining sample becomes less representative [107]. In cognitive studies, participants who remain are often characterized by better levels of education and health, potentially skewing the results to show less cognitive decline [108]. To minimize attrition, it is important to make all study participants feel valued as “citizen scientists” [109] and not merely research subjects. As shown in the Rush RCT [2], equal time, attention, and engagement must be given to participants in all study arms so that there is equipoise between them.

#### 4.2.7. Statistical Power and Sample Size

Ensure the study is powered to detect the meaningful minimum difference that has been identified as the study’s primary outcome. In the Rush RCT [2], the study is powered to test the effect of the MIND intervention over the control intervention based on the annual rate of change in the global cognitive score (a composite of the 12 neuropsychological tests). Based on a sample size of 300 participants in each of the two groups, the MIND trial has >85% power to detect a between-group difference of 2% in the annual rate of cognitive decline (two-sided *p* = 0.05) considering a possible 5% drop-out at each visit.

#### 4.2.8. Additional Recommendations

Begin testing populations at an earlier age. A potential limitation of the Rush RCT [2] is its focus on older adults aged 65–84 years. While the study of dementia in older adults is the norm, the recommendation of the Alzheimer’s Association and the NAS is to begin studies at an earlier age, since changes in brain physiology and cognitive abilities may occur years before symptoms appear, and lifestyle interventions may have a more significant impact if started at an earlier life-stage [3,58] (pp. 81–82). The challenge of studying a younger population is that there is less cognitive decline, so these populations need to be studied for a longer duration to determine if dietary patterns have an effect.

Include the intake of anticholinergic drugs, including antihistamines, and benzodiazepines as potential confounders of study outcomes. These drugs are associated with adverse cognitive effects in older adults [110,111] and may lead to spurious results if they are not analyzed as part of the study. Only one RCT analyzed [34] included anticholinergic drugs as a potential confounder.

## 5. Conclusions

Alzheimer’s disease is a critical public health issue given the societal, financial, individual, and family burden. Projected numbers in the next three decades may increase beyond the current staggering estimates based on the unknown and potentially lasting effects of social isolation due to COVID-19. The traditional pharmaceutical approach has not demonstrated a cure or reduction in symptoms over the long term, while lifestyle approaches, including dietary patterns, provide a low-risk, low-cost alternative that may dramatically change the trajectory of this disease.

More RCTs are needed so that dietary patterns with an evidence base for effectiveness can be identified and can reach a broader population more quickly. The design of future RCTs should address gaps and challenges that may have led to inconsistent study findings in the past including inaccuracy in assessing food intake, multiple dietary scoring systems, a shifting metric among studies focused on the Mediterranean diet, a lack of standards in the tools used to assess cognitive decline, and studies that were underpowered or follow-up periods that were too short to detect cognitive change. Because lifestyle interventions such as diet may have greater impact at earlier life stages, studies should begin to include younger populations.

It is vital to accelerate the study of these neuroprotective diets through explanatory and pragmatic trials to potentially mitigate the devastating effects of a disease that is projected to affect more than 100 million people in the next 30 years. 

## Figures and Tables

**Figure 1 nutrients-13-02264-f001:**
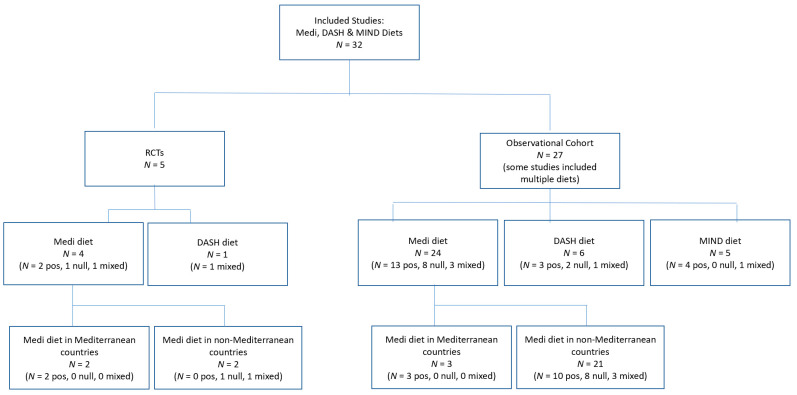
Overview of included studies.

**Table 1 nutrients-13-02264-t001:** Medi/DASH/MIND dietary components and potential neuroprotective benefits.

Dietary Component	Vascular Protection	Anti-Inflammatory Protection	Antioxidant Protection	Neural Protection
Whole Grains [2,13,29,41,42,43,44,45,46]	X	X	X	X
Vegetables (MIND diet emphasizes leafy greens in addition to other vegetables) [2,29,39,41,42,43,44,45,46,47]	X	X	X	
Fruits (MIND diet emphasizes berries over other fruits) [2,29,39,41,42,43,44,45,46,47]	X	X	X	
Fatty Fish (e.g., salmon, tuna, sardines) [2,13,14,31,41,42]	X	X	X	X
Legumes/Beans [2,41,45,48]	X	X	X	X
Nuts [2,29,41,46,47]	X	X	X	X
Olive Oil (Medi, MIND diets) [33,49,50]		X	X	
Red Wine (Medi, MIND diets) [29,46,49,51,52,53]		X	X	

“X” indicates association with the neuroprotective benefit listed.

**Table 2 nutrients-13-02264-t002:** Comparison of Mediterranean, DASH, and MIND dietary components. The components in bold are common across all three dietary patterns.

	Mediterranean [17,54,55]	DASH [56,57]	MIND [17]
**HIGH**	Olive Oil (exclusive)	-	-
**AMOUNTS**	**Fish**	-	-
	**Grains** (non-refined)	**Grains**	**Grains** (whole)
	Fruits	Fruits	Berries
	Vegetables	**Vegetables**	Green leafy vegetables
	-	**-**	**Other vegetables**

	Potatoes	**-**	**-**
	Legumes	Legumes	**-**
	Beans	**-**	Beans
	**Nuts**	**Nuts**	**Nuts**
	Seeds	Seeds	-
	-	Low-fat diary	-
**MODERATE**	**Poultry**	**Poultry**	**Poultry**
**AMOUNTS**	-	**Fish**	**Fish**
	Alcohol	-	Alcohol
	-	-	Olive oil (primary)
	Full-fat dairy	-	-
**SMALL**	**Red meat**	Red meat	**Red meat**
**AMOUNTS**	Processed meat	-	-
	**Sweets**	**Sweets**	**Sweets/pastries**
	-	Saturated fat as a	-
		% of total fat	
	-	Sodium <2400 mg/d	**-**
**RESTRICTED**	-	-	Cheese
	-	-	Butter/margarine
	-	-	Fast/fried foods

**Table 3 nutrients-13-02264-t003:** RCTs and observational studies in this review.

Type	Author, Study Name	Year #Participants Country Follow-Up	Dietary Pattern	Purpose	Findings	Outcome	Dietary Intake	Dietary Pattern Scoring	Cognitive Scoring
RCT	Martinez-Lapiscina et al. PREDIMED NAVARA [33]	2013 *N* = 522 Spain 6.5 years	Medi	Compared the effects of two interventions of a Mediterranean diet vs. a low-fat diet on cognition or incidences of dementia. One intervention was supplemented with extra virgin olive oil (EVOO) and one supplemented with nuts.	Participants in both Medi Diet groups showed higher mean global cognitive function scores vs. the control group, with more significant differences in the Medi Diet + EVOO group. Medi Diet + EVOO group showed mean global cognitive function scores with significant differences versus the control group (*p* = 0.005 for MMSE, *p* = 0.001 for CDT). MedDiet + Nuts group versus the control group showed mean global cognitive differences vs. the control group (*p* = 0.015 for MMSE and *p* = 0.048 for CDT).	Positive correlation	FFQ	14-item Medi diet questionnaire	MMSE + CDT
RCT	Valls-Pedret et al. PREDIMED-NAVARRA [34]	2015 *N* = 447 Spain 4.1 years	Medi	Compared the effects of two interventions of a Mediterranean diet vs. a low-fat diet on cognition or incidences of dementia. One intervention was supplemented with extra virgin olive oil (EVOO) and one supplemented with nuts.	Both the Mediterranean diet plus olive oil and the Mediterranean diet plus nuts demonstrated improved composite measures of cognitive function, although the results varied based on the cognitive assessment. Similarly adjusted cognitive composites (mean z scores with 95% CIs) for changes above baseline of the memory composite were 0.04 (−0.10 to 0.17) for the Mediterranean diet plus olive oil; 0.10 (−0.04 to 0.24; *p* = 0.04 vs. controls) for the Mediterranean diet plus nuts; and −0.16 (−0.32 to −0.01) for the control diet. Respective changes from baseline of the frontal cognition composite were 0.23 (0.02 to 0.43; *p* = 0.004 vs. controls); 0.03 (−0.26 to 0.32); and −0.33 (−0.57 to −0.09). Changes from baseline of the global cognition composite were 0.04 (−0.12 to 0.20; *p* = 0.008 vs. controls) for the Mediterranean diet plus olive oil; −0.04 (−0.27 to 0.19) for the Mediterranean diet plus nuts; and −0.37 (−0.56 to −0.17) for the control diet. All cognitive composites significantly (*p* < 0.05) decreased from baseline in controls.	Positive correlation	FFQ	14-item Medi diet questionnaire	Battery
RCT	Knight et al. MedLey Study [60]	2016 *N* = 137 Australia 0.5 years	Medi	Compared an intervention arm (Medi Diet) vs. a control arm (habitual diet) on cognition and cognitive decline.	No effect of Medi diet vs. habitual diet on cognition. No difference in executive function (*p* = 0.33); speed of processing (*p* = 0.15); memory (*p* = 0.50); visual-spatial ability (*p* = 0.48); or overall age-related cognitive performance (*p* = 0.19).	Null	FFQ	An Australian version based on the food content of 15 Medi diets and the nutrient content of 8 Medi diets	Battery
RCT	Wardle et al. (No Study Name) [61]	2000 *N* = 176 UK 0.2 years	Medi	Compared adults with elevated serum cholesterol levels assigned to one of two intervention groups (low-fat diet or Mediterranean diet) and a control group (wait-list control) to determine effects of diet on serum cholesterol and cognitive function.	There were no between-group differences on three measures of cognitive function, but for a fourth measure, which involved the task with the greatest processing load, the two intervention groups did significantly worse (*p* < 0.001) than the control group. The change in performance was correlated with the change in total serum cholesterol level (*p* = 0.01).	Mixed findings	Dietary diaries	Omitted, except for guidelines to increase vegetables, fruits, and oily fish; reduce dietary fat to 30% of energy; substitute predominantly mono-unsaturated fats for saturated fats	Battery
RCT	Smith et al. ENCORE [62]	2010 *N* = 124 0.3 years US	DASH	Compared two intervention arms (DASH diet only; DASH diet + weight management) vs. a control arm to determine effects on executive-function-memory-learning (EFML).	DASH diet alone participants showed no improvement in EFML, but better psychomotor speed (*p* < 0.036) than the usual diet control. DASH diet combined with behavioral weight management program showed greater improvements in EFML (*p* < 0.008) and psychomotor speed (*p* < 0.023) than the control.	Mixed findings	Not stated (but measured change against the Healthy Eating Index)	Omitted	Battery
Obs	Berendsen Nurses Health Study [38]	2017 *N* = 16,144 US Years = 6.0	DASH	Cognition and cognitive decline	Greater adherence to long-term DASH score was associated with better average cognitive function (*p* = 0.009 for global cognition) but not associated with change in cognitive function over 6 years.	Mixed findings	FFQ	Med 0–9	Battery
Obs	Berendsen Nurses Health Study [63]	2017 *N* = 16,058 US 6.0 years	MIND	Cognition and cognitive decline	Greater adherence to MIND diet was positive for verbal memory score in extreme quintiles (*p* = 0.006), but not with cognitive decline over 6 years in global cognition, verbal memory, or TICS score.	Mixed findings	FFQ	MIND 0–15	Battery
Obs	Bhushan Health Professionals’ Follow-Up Study [64]	2017 *N* = 27,842 US 16.0 years	Medi	Cognition and cognitive decline	Long-term adherence to Medi diet was strongly related to lower subjective cognitive decline as measured by a 6-question Subjective Cognitive Function score (SCF). Compared with men having a Medi diet score in the lowest quintile, those in the highest quintile had 36% lower odds of a poor SCF score (*p* < 0.001) and 24% lower odds of a moderate SCF score (*p* < 0.001).	Positive correlation	FFQ	Med 0–9	SCF
Obs	Cherbuin PATH Through Life study [65]	2012 *N* = 1528 Australia 4.0 years	Medi	Cognitive decline, incidence of mild cognitive impairment (MCI) or dementia	There were no protective benefits of Medi diet. Instead, a high intake of mono-unsaturated fatty acids and excessive caloric intake were predictive of MCI (*p* < 0.01).	Null	FFQ	Med 0–9	Clinical Dementia Rating battery
Obs	Galbete et al. SUN [66]	2015 *N* = 823 Spain 2.0 years	Medi	Cognition and cognitive decline	Higher cognitive decline was observed in participants with low to moderate adherence to the Medi diet than among those with better adherence (adjusted difference = −0.56 points in the Telephone Interview for Cognitive Status-modified (TICS-m) score, 95% CI = −0.99 − 0.13).	Positive correlation	FFQ	Med 0–9	TICs-m battery
Obs	Gardener et al. AIBL [67]	2012 *N* = 970 Australia 1.5 years	Medi	Cognition and cognitive decline within healthy control (HC) subjects, those with mild cognitive impairment (MCI), and those with Alzheimer’s Disease (AD)	Lower adherence to the Medi diet was associated with MCI and AD. Compared with HC subjects, subjects with AD had a lower mean Medi diet score (*p* < 0.001); and compared with HC subjects, subjects with MCI had a lower mean Medi diet score (*p* < 0.05). Compared with the reference HC category, each additional unit in the Medi diet score was associated with 13–19% lower odds of being in the MCI category, and 19–26% lower odds of being in the AD category.	Positive correlation	FFQ	Med 0–9 modified for Australian population	MMSE
Obs	Gardener et al. AIBL [68]	2015 *N* = 527 Australia 3.0 years	Medi	Cognition change based on three dietary patterns: AusMeDi (modified Mediterranean); Western diet; Prudent diet	Results for global cognitive score and visuospatial functioning not significant. In a subset of APOE ε4 allele carriers, higher adherence to AusMeDi diet was associated with better performance in executive function in (*p* < 0.01).	Null	FFQ	Med 0–9 modified for Australian population	Battery
Obs	Gu et al. WHICAP II [69]	2010 *N* = 1219 US 4.0 years	Medi	Effect of adherence to the Mediterranean diet (MeDi) and Alzheimer’s disease (AD)	Better adherence to Medi was significantly associated with lower risk for AD. Compared to those in the lowest tertile of Medi, subjects in the highest tertile had 34% less risk of developing AD (p-for-trend = 0.04)	Positive correlation	FFQ	Med 0–9	Battery
Obs	Haring et al. WHI Memory study [70]	2016 *N* = 6425 US 9.11 years	Medi DASH	Cognitive decline, incidence of mild cognitive impairment (MCI) or Parkinson’s Disease (PD)	Medi and DASH dietary patterns were not associated with cognitive decline in older women (*p* = 0.30 for Medi diet, *p* = 0.23 for DASH diet). Adherence to these healthy dietary patterns did not modify the risk for cognitive decline in women with hypertension.	Null: Med Null: DASH	FFQ	Med 0–9	Battery
Obs	Hosking et al. PATH Through Life study [40]	2019 *N* = 1220 Australia 12.0 years	Medi MIND	Cognitive impairment (MCI or dementia)	In adjusted logistic regression models, the MIND diet (OR = 0.47, 95% CI 0.24, 0.91), but not the Medi diet, was associated with reduced odds of 12-year cognitive impairment. For every 1-point increase in MIND score, the odds of impairment decreased by 19%. There were no associations between Medi and incidence of MCI/dementia.	Pos: MIND Null: Med	FFQ	Med 0–9 Med 0–55 MIND 0–15	Battery
Obs	Kesse-Guyot et al. SUNVIMAX [71]	2013 *N* = 3083 France 2.0 years	Medi	Cognition	Adherence to Medi did not provide a beneficial effect on cognitive function. However, within the battery of six neuropsychological tests, a lower phonemic fluency score (*p* = 0.048) and lower backward digit span score (*p* = 0.03) were observed with decreasing Medi score.	Mixed findings	24HR	Med 0–9	Battery
Obs	Koyama et al. Health ABC Study [72]	2015 *N* = 2326 US 7.9 years	Medi	Cognition and cognitive decline	Adherence to Medi shows mixed results between Black and white participants. In Black participants, high Medi scores had lower mean rate of cognitive decline on the 3MS score, which measures several cognitive domains (*p* = 0.01). No similar association was found in white participants (*p* = 0.14). Mechanisms for differences undetermined.	Mixed findings	FFQ	Med 0–9	3MS
Obs	Morris et al. MAP [39]	2015 *N* = 923 US 4.5 years	Medi DASH MIND	Cognition and cognitive decline	High adherence to all three diets demonstrated a reduced risk of AD. Moderate adherence to the MIND diet only also showed decreased AD risk. In adjusted proportional hazards models, the second (HR = 0.65, 95% CI 0.44, 0.98) and highest tertiles (HR = 0.47, 95% CI 0.26, 0.76) of MIND diet scores had lower rates of AD versus tertile 1, whereas only the third tertiles of the DASH (HR = 0.61, 95% CI 0.38, 0.97) and Mediterranean (HR = 0.46, 95% CI 0.26, 0.79) diets were associated with lower AD rates.	Pos: Med Pos: DASH Pos: MIND	FFQ	Med 0–55 Dash 0–10 MIND 15	Battery
Obs	Morris et al. MAP [17]	2015 *N* = 960 US 4.7 years	MIND	Cognition and cognitive decline	In adjusted mixed models, the MIND score was positively associated with slower decline in global cognitive score (*p* < 0.0001) and with each of the five cognitive domains. The difference in decline rates for the top tertile of MIND diet scores vs. the lowest was the equivalent of being 7.5 years younger in age.	Positive correlation	FFQ	MIND 0–15	Battery
Obs	Olsson et al. Sweden ULSAM [73]	2015 *N* = 1038 Sweden 12.0 years	Medi	Cognitive decline rates based on changes to the Mini-Mental State Examination (MMSE) score for participants who ate a Swedish Modified Medi Diet (mMDS); the WHO-recommended Healthy Diet Indicator; or a low carbohydrate high protein diet (LCHP)	There were no strong associations with development of cognitive dysfunction for any of the dietary patterns investigated. Neither the continuous Medi score nor the higher tertile of the Medi score was associated with lower risk of AD, dementia, or cognitive impairment. In a secondary analysis, in a subset of participants based on energy intake according to the Goldberg cut-off, the highest tertile of the Medi score was significantly associated with a lower risk of cognitive impairment (OR: 0.32, 95% CI: 0.11, 0.89).	Null	24HR	Med 0–9 modified for the Swedish population	MMSE
Obs	Qin China Health and Nutrition Survey [37]	2015 *N* = 1650 China 7.0 years	Medi	Cognition and cognitive decline	Adherence to the Medi diet showed different outcomes based on age group. For participants =/>65, quintile 3 of the Chinese adapted Medi vs. quintile 1 had a slower rate of cognitive decline (β = 0.042). There was no association in adults <65.	Mixed findings	24HR	Med 0–9	Battery
Obs	Roberts et al. [No Study Name] [74]	2010 *N* = 1223 US 2.2 years	Medi	Mild Cognitive Impairment (MCI)	Higher Medi score tertile was not significantly associated with reduced risk of MCI (*p* = 0.24).	Null	FFQ	Med 0–9	Battery
Obs	Samieri et al. Nurses Health Study [75]	2013 *N* = 16,058 US 13.0 years	Medi	Cognition and cognitive decline	Medi was not associated with decline in global cognition or verbal memory. In a secondary approach analyzing a subset of participants age 70+, each higher quintile of long-term Medi score was linearly associated with better multivariable-adjusted mean cognitive scores (*p* = 0.004, 0.002, and <0.001 for Telephone Interview for Cognitive Status (TICS), global cognition, and verbal memory, respectively.)	Null	FFQ	Med 0–9 modified for US population	Battery
Obs	Shakersain et al. SNAC-K [76]	2018 *N* = 2223 Sweden 6.0 years	Medi DASH MIND	Analyze a change in global cognition based on the Mini-Mental State Examination (MMSE) when comparing the Nordic Prudent Dietary Pattern (NPDP) to the Medi, MIND, DASH, and Baltic Sea Diets (BSD).	Moderate (*p* < 0.019) and high (*p* < 0.001) adherence to MIND diet was related to less cognitive decline as measured by the MMSE. Only high adherence (0.002) to Medi was related to less cognitive decline. No associations were observed between DASH and cognitive decline. Moderate (*p* < 0.001) and high (*p* < 0.001) adherence to the NPDP were related to less cognitive decline. No associations were observed between BSD and cognitive decline.	Med: Pos DASH: Null MIND: Pos	FFQ	Med 0–55 Dash 8–40 modified for the Swedish population	MMSE
Obs	Tanaka et al. InCHIANTI [77]	2018 *N* = 832 Italy 10.1 years	Medi	Global cognition measured by the Mini-Mental State Examination (MMSE)	Adherence to Medi was associated with a lower risk of cognitive decline as measured by the MMSE. Those in the highest adherence group (OR = 0.48, 95% CI: 0.29–0.79) and medium adherence group (OR = 0.64, 95% CI: 0.41–0.99) were less likely to experience cognitive decline. The annual average decline in MMSE scores was 0.4 units; for those in the high and medium adherence group, this decline was attenuated by 0.34 units (*p* < 0.001) and 0.16 units (*p* = 0.03), respectively.	Positive correlation	FFQ	Med 0–9	MMSE
Obs	Tangney et al. Chicago Health and Aging Project [78]	2011 *N* = 3790 US 7.6 years	Medi	Cognition and cognitive decline	Higher Medi scores were associated with reduced decline in cognitive function (*p* = 0.0004). Higher Medi diet wine scores were also associated with reduced decline in cognitive function (*p* = 0.0009).	Positive correlation	FFQ	Med 0–55	Battery
Obs	Tangney et al. MAP [79]	2014 *N* = 826 US 4.1 years	Medi DASH	Cognition and cognitive decline	Both the DASH and Mediterranean diet patterns were associated with slower rates of cognitive decline in the same cohort of older persons. A 1-unit difference in the DASH score was associated with a slower rate of global cognitive decline by 0.007 standardized units (standard error of estimate = 0.003, *p* < 0.03). A 1-unit-higher Medi score was associated with a slower rate of global cognitive decline by 0.002 standardized units (standard error of estimate = 0.001, *p* < 0.01).	Pos: Med Pos: DASH	FFQ	Med 0–55	Battery
Obs	Titova et al. PIVUS [80]	2013 *N* = 194 Sweden 5 years	Medi	Association between dietary habits, cognitive functioning and brain volumes in older adults	Low consumption of meat and meat products was linked to a better performance on the seven-minute screening (7MS) test (*p* = 0.001) and greater total brain volume (i.e., the sum of white and gray matter, *p* = 0.03) when controlling for potential confounders (e.g., BMI). Integrating all dietary features into the total modified Medi diet score explained less variance in cognitive functioning and brain volumes than the single dietary component meat intake.	Positive correlation	7-day food diary	Med 0-8 (adapted from Med 0–9 for the Swedish population)	7MS score
Obs	Trichopoulou et al. EPIC-Greece [32]	2015 *N* = 401 Greece 7.0 years	Medi	Cognition and cognitive decline	Decline in cognition measured by MMSE was inversely associated with adherence to Medi, with only vegetable consumption exhibiting a substantial inverse effect. For mild vs. no decline, *p* = 0.012. For substantial vs. no decline, *p* = 0.025.	Positive correlation	FFQ	Med 0–9	MMSE
Obs	Tsivgoulis et al. REGARDS [81]	2012 *N* = 17,148 US 4+/−1.5	Medi	Cognition, Incident Cognitive Impairment (ICI)	Higher adherence to the Medi diet was associated with lower likelihood of ICI before (OR = 0.89; 95% CI 0.79–1.00) and after adjustment for potential confounders (OR = 0.87; 95% CI 0.76–1.00). Higher adherence was associated with lower likelihood of ICI in non-diabetic (*p* = 0.0066) but not diabetic (*p* = 0.1063) individuals; diabetics were less adherent to the diet.	Positive correlation	FFQ	Med 0–9	Six-item screener
Obs	Vercambre et al. WACS [82]	2012 *N* = 2504 US 5.4 years	Medi	Cognitive decline	In both basic- and multivariable-adjusted models, consuming a Mediterranean style diet was not related to cognitive decline. The mean multivariable-adjusted difference 95% CI in rates of change in the global composite score was 0.01 (−0.01, 0.02) between the second and first tertile of the Medi score, and 0.00 (−0.02, 0.01) between the top tertile and first tertile (*p* = 0.88).	Null	FFQ	Med 0–9	Battery
Obs	Wengreen et al. Cache County Study on Memory, Health, and Aging [35]	2013 *N* = 3831 US 11.0 years	MediDASH	Global cognition	Higher DASH and Mediterranean diet scores were associated with higher average cognition measured by Modified Mini-Mental State (3MS) test scores over an 11-year period. Participants in quintile 5 of DASH averaged 0.97 points higher than those in quintile 1 (*p* = 0.001). The corresponding difference for Mediterranean quintiles was 0.94 (*p* = 0.001).	Pos: MediPos: DASH	FFQ	Med 0–9 (modified for low alcohol consumption in population) Dash 8–40	3MS Battery

Abbreviations: RCT, randomized clinical trial; Obs, observational trial; FFQ, Food Frequency Questionnaire; 24HR, 24 Hour Dietary Recall; Medi, Mediterranean diet; EVOO, extra virgin olive oil; MMSE, Mini-Mental State Examination; CDT, Clock Drawing Test; SCF, Subjective Cognitive Function; TICS, Telephone Interview For Cognitive Status; 7MS, Seven-Minute Screening; 3MS, Modified Mini-Mental State Test.

**Table 4 nutrients-13-02264-t004:** Representative examples of tests used in a neuropsychological test battery [2,98].

Purpose/Cognitive Domain	Representative Cognitive Tests
**Screening**	Mini-Mental State Examination (MMSE)
	Montreal Cognitive Assessment (MCOA)
**Executive Functioning**	Stroop Test
	Initial Letter Fluency (ILF)
	Excluded Letter Fluency (ELF)
	Tower of London (TOL)
	Trail Making Test-Part B
	NIH Toolbox Flanker Test
**Episodic Memory**	Rey Auditory Verbal Learning Test (RALVT)
	Letter-Number Sequencing Subtest from the Wechsler Adult Intelligence Scale (WAIS-IV)
	Immediate and Delayed Recall of the East Boston Story
	Word List Learning, Recall and Recognition from the Consortium to Establish a Registry for Alzheimer’s Disease (CERAD)
**Semantic Memory**	Category Fluency (fruits, animals)
	Multilingual Naming Test
**Working Memory**	Digit Span Forward (DSF)
	Digit Span Backward (DSB)
**Perceptual Speed**	Symbol Digit Modalities Test
	Number Comparison
	Stroop Word Reading
	Stroop Color Naming
	Symbol Search and Coding Core Subtests from WAIS IV
	Trail Making Test-Part A
**Visuospatial Ability**	Judgment of Line Orientation
	Standard Progressive Matrices
	Benton Visual Retention Test (BVRT)

## Data Availability

The data that support the findings of this study are available from the corresponding author (S.C.D.), upon reasonable request.

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
