# Peer review of "A Critical Review of the Study of Neuroprotective Diets to Reduce Cognitive Decline"

_nutrients, 2021, doi:10.3390/nu13072264_

Round 1
Reviewer 1 Report
The article is very extensive but interesting, and the subject is important because dementia is a major public health problem as it has serious family, social and financial consequences. The development of reliable, scientifically based guidelines on the nutrition of people at risk of dementia may provide an opportunity to relieve not only the family but also many sectors of the State. Suggestion: In the chapter "Beyond Study Limitations to Practical Recommendations", I would also suggest the need to analyze the drugs taken by the participants of the study, because some geriatric pharmaceutical preparations may have an effect on the impairment or improvement of cognitive functions.
Author Response
Your Comment: The article is very extensive but interesting, and the subject is important because dementia is a major public health problem as it has serious family, social and financial consequences. The development of reliable, scientifically based guidelines on the nutrition of people at risk of dementia may provide an opportunity to relieve not only the family but also many sectors of the State.
Our Response: Thank you for your positive comments about our article.
Your Suggestion: In the chapter "Beyond Study Limitations to Practical Recommendations", I would also suggest the need to analyze the drugs taken by the participants of the study, because some geriatric pharmaceutical preparations may have an effect on the impairment or improvement of cognitive functions
Our Response: This is an excellent suggestion. We interpreted this feedback to mean that future RCTs should analyze certain types of drugs, such as benzodiazepines and anticholinergic drugs, as possible confounders since they are associated with adverse cognitive effects and may lead to spurious study results. The effects have been documented by the National Academy of Medicine and the American Geriatrics Society. We did not include pharmaceutical preparations that may improve cognitive function because the current literature did not demonstrate a strong relationship between drugs like cholinesterase inhibitors and memantine on cognitive test scores or the study populations were very small or specialized where relationships were shown.
Below is the paragraph we added to section 5.2.8 Additional Recommendations:
Include the intake of anticholinergic drugs, including antihistamines, and benzodiazepines as potential confounders of study outcomes. These drugs are associated with adverse cognitive effects in older adults [108, 109] and may lead to spurious results if they are not analyzed as part of the study. Only one RCT we analyzed [23] included anticholinergic drugs as a potential confounder.
Reviewer 2 Report
This is a comprehensive, didactic review of diet and ADRD/brain health. The review has a number of strengths, however, there are also limitations including:
- the literature search strategy did not include the word 'diet'
- several highly didactic sections are included, which are unexpected in a research hypothesis-driven review. These didactic sections are not introduced from the onset of the review. For example, there a much detail provided about dietary assessment and nutritional epidemiology. While interesting, and known as a discipline, its extensive role in this review is not understood.
- Please take caution regarding references to clinical AD versus neuropathological AD. These concepts are mixed in this review.
- In general, this review can be greatly shortened and focused. One can get lost in all of the detail, which does not focus the research question.
- Some of the comments are opinions, not cited, e.g., the length of a diet and cognition/aging study should be a minimum of 3 years. Details related to RCT planning are based on the age of participants, as well as their baseline cognitive status, and often on biomarker status, whether brain imaging or fluid biomarker levels.
Comments in the .pdf are attached.

Author Response
Thank you for your comments and suggestions. Addressing these has helped make this a stronger and more focused review. Our responses are provided below in purple text. We have also attached a Word document with additional edits using track changes.
Comment: The literature search strategy did not include the word 'diet'
Response: To narrow the scope of this review, the focus was on the three diets commonly associated with neuroprotective benefits. Adding the word “diet” to the search terms would have brought in a small number of additional studies that would have been excluded, by our inclusion/exclusion criteria. Our approach was a bit more exhaustive since it allowed for the identification of studies that might have listed their interventions as Mediterranean, DASH or MIND “eating patterns” or “dietary patterns.” Additionally, a key approach to determining if most or all of the relevant studies were identified was the careful review of the three primary meta-analyses that were identified, and the list of studies they cited. If our search did fail to identify relevant studies, the implications for the findings and conclusions should have been very minor.
Comment: Several highly didactic sections are included, which are unexpected in a research hypothesis-driven review. These didactic sections are not introduced from the onset of the review. For example, there a much detail provided about dietary assessment and nutritional epidemiology. While interesting, and known as a discipline, its extensive role in this review is not understood.
Response: We appreciate this feedback. We removed an entire section, “The Biology of Potentially Neuroprotective Dietary Components.” We included relevant information from this section in two sentences within the introduction and added a new Table 2 to summarize the content.
We also shortened the “Dietary Intake Measures” section to two paragraphs by removing explanatory text. We kept content regarding limitations in the tools that could lead to inconsistent study outcomes which is relevant to this paper.
Comment: Please take caution regarding references to clinical AD versus neuropathological AD. These concepts are mixed in this review.
Response: This is an excellent point. The most confusing section appeared to begin at line 122 when we talked about measures to diagnose CATD, which is the clinical Alzheimer’s type dementia. We removed this phrase and replaced it with AD.
Comment: In general, this review can be greatly shortened and focused. One can get lost in all the detail, which does not focus the research question.
Response: This feedback appears connected to the earlier point about long didactic sections. We removed and/or greatly shortened these sections to ensure the content was focused on the research question.
Comment: Some of the comments are opinions, not cited, e.g., the length of a diet and cognition/aging study should be a minimum of 3 years. Details related to RCT planning are based on the age of participants, as well as their baseline cognitive status, and often on biomarker status, whether brain imaging or fluid biomarker levels.
Response: We modified the “Length of Follow-Up” section beginning at Line 578 to reflect only cited references.
Additional changes and comments can be found in the attached.
Response: We appreciate the detailed feedback including comments on abbreviations and minor word changes. All comments from the PDF have been addressed and are noted through track changes as well as Comments.
